# Dynamics of Organic Acids during the Droplet-Vitrification Cryopreservation Procedure Can Be a Signature of Oxidative Stress in *Pogostemon yatabeanus*

**DOI:** 10.3390/plants12193489

**Published:** 2023-10-06

**Authors:** Hyoeun Lee, Byeongchan Choi, Songjin Oh, Hana Park, Elena Popova, Man-Jeong Paik, Haenghoon Kim

**Affiliations:** 1Department of Agricultural Life Science, Sunchon National University, Suncheon 57922, Republic of Korea; yee0430@scnu.ac.kr (H.L.); qkrgksk1102@naver.com (H.P.); 2College of Pharmacy, Sunchon National University, Suncheon 57922, Republic of Korea; chlqudcks456@naver.com (B.C.); osj7797@naver.com (S.O.); 3K.A. Timiryazev Institute of Plant Physiology of Russian Academy of Sciences, Botanicheskaya 35, Moscow 127276, Russia; elena_aygol@hotmail.com

**Keywords:** alternative vitrification solutions, A3-80%, cryopreservation protocol optimization, endangered wild species, metabolic profiling, oxidative stress, regrowth medium

## Abstract

Cryopreservation in liquid nitrogen (LN, −196 °C) is a unique option for the long-term conservation of threatened plant species with non-orthodox or limitedly available seeds. In previous studies, a systematic approach was used to develop a droplet-vitrification (DV) cryopreservation protocol for *Postemon yatabeanus* shoot tips that includes preculture with 10% sucrose, osmoprotection with C4-35%, cryoprotection with A3-80% vitrification solution, and a three-step regrowth starting with the ammonium-free medium. The tricarboxylic acid (TCA) cycle is a crucial component of plant cell metabolism as it is involved in redox state regulation and energy provision. We hypothesized that organic acids (OAs) associated with the TCA and its side reactions indirectly indicate metabolism intensity and oxidative stress development in shoot tips under the cryopreservation procedure. In this study, the contents of 14 OAs were analyzed using gas chromatography–tandem mass spectrometry (GC-MS/MS) in *P. yatabeanus* shoot tips in a series of treatments including individual steps of the DV procedure, additional stress imposed by non-optimum protocol conditions (no preculture, no osmoprotection, various vitrification solution composition, using vials instead of aluminum foils, etc.) and regrowth on different media with or without ammonium or growth regulators. The possible relation of OA content with the total cryoprotectant (CPA) concentration and shoot tips regeneration percentage was also explored. Regeneration of cryopreserved shoot tips reduced in descending order as follows: standard protocol condition (91%) > non-optimum vitrification solution (ca. 68%) > non-optimum preculture (60–62%) > regrowth medium (40–64%) > no osmoprotection, cryopreservation in vials (28–30%). Five OAs (glycolic, malic, citric, malonic, and lactic) were the most abundant in the fresh (control) shoot tips. The dynamic pattern of OAs during the DV procedure highly correlated (*r* = 0.951) with the total CPA concentration employed: it gradually increased through the preculture, osmoprotection, and cryoprotection, peaked at cooling/rewarming (6.38-fold above control level), and returned to the fresh control level after 5 days of regrowth (0.89-fold). The contents of four OAs (2-hydroxybutyric, 3-hydroxypropionic, lactic, and glycolic) showed the most significant (10-209-fold) increase at the cooling/rewarming step. Lactic and glycolic acids were the major OAs at cooling/rewarming, accounting for 81% of the total OAs content. The OAs were categorized into three groups based on their dynamics during the cryopreservation protocol, and these groups were differently affected by protocol step modifications. However, there was no straightforward relationship between the dynamics of OAs and shoot tip regeneration. The results suggest that active modulation of OAs metabolism may help shoot tips to cope with osmotic stress and the chemical cytotoxicity\ of CPAs. Further intensive studies are needed to investigate the effect of cryopreservation on cell primarily metabolism and identify oxidative stress-related biomarkers in plant materials.

## 1. Introduction

Cryopreservation, the storage of biological material in liquid nitrogen (LN, −196 °C), enables the long-term preservation of genetic resources, exceptionally short-lived seeds, vegetatively propagated species, and seed-producing heterozygous species [1]. Droplet-vitrification (DV) method, which is frequently employed for cryopreserving plant materials, is a multi-stage procedure with many factors involved from stage (1) material preparation to (2) pre-LN (preculture, osmoprotection, and cryoprotection with vitrification solution), (3) LN (cooling, rewarming, and unloading), and (4) post-LN (regrowth). During the cryopreservation procedure, plant cells and tissues are vulnerable to intracellular ice crystallization and extensive cellular dehydration during cooling and rewarming (C/W) [2] and osmotic stress and chemical toxicity during cryoprotection with highly concentrated plant vitrification solutions (PVS) [3]. The reactive oxygen species (ROS)-induced oxidative stress [4,5,6], typically maximized during PVS treatment and C/W, also decreases the recovery of the cryoprotected-control (LNC) and cryopreserved (LN) shoot tips. Mitochondrial function providing adenosine triphosphate (ATP) is critical in recovering cryopreserved explants; however, it may be significantly reduced after cryopreservation [7]. For example, over 50% of metabolic rates were decreased in the shoot tips of two *Anigozanthos* species after cryopreservation [6].

*Pogostemon yatabeanus* (Makino) Press [8] is an endemic and endangered species in Korea, which limitedly occurs in wetlands [9]. Earlier, we developed an in vitro culture system for this species and cryopreservation of the shoot tips of in vitro-propagated plants. The post-cryopreservation (LN) regeneration increased from 20% to 90% via the vigorous growth of donor plants, cryoprotection with alternative PVS A3-80%, followed by a three-step regrowth initially with ammonium-free medium with growth regulators [10,11,12]. Even with the optimized procedure, cryopreserved shoot tips showed slower regrowth and lower regeneration than non-treated fresh control.

Since energy provision is critical for the recovery of cryopreserved materials, metabolomics studies during cryopreservation have recently increased [7,13]. Metabolomics involves a comprehensive analysis of small molecules, such as organic acids (OAs), amino acids, peptides, fatty acids, and carbohydrates, which provide physiological information about various metabolic pathways in cells, tissues, and organs [14,15,16]. For example, such analysis helped reveal plant response to growth-promoting endophytic bacteria [17]. However, metabolomic analysis of cryopreserved plant materials has rarely been performed. Mitochondria are related to energy metabolism in the tricarboxylic acid (TCA) cycle, which was shown to be adversely affecteds by cryopreservation [18]. The metabolic rate of cryopreserved shoot tips reduced significantly and did not increase within two days [18]. Stress tolerance responses may include nutrient acquisition and detoxifying toxic minerals, such as aluminum, pH, and redox equilibrium, to avoid cell ROS accumulation [19]. Sugars, amino acids, and OAs accumulate under abiotic stress [20]. Hence, analysis of these metabolites can be essential to link metabolic changes with plant responses during cryopreservation.

In this study, we performed the profiling of 14 organic acids, including TCA cycle-related OAs, as the first indicator of altered energy metabolism in *P. yatabeanus* shoot tips during their cryopreservation via the DV procedure. We investigated the impact of various factors, including individual steps of the cryopreservation procedure, composition of vitrification solution, and regrowth strategy on the content of individual OAs. Additional stress imposed by non-optimum pre-LN conditions was also studied for its effect on OA metabolism. To the best of our knowledge, this is the first extensive investigation of the OA content in shoot tips of the endangered plant species during cryopreservation.

## 2. Results

### 2.1. Effect of Droplet-Vitrification Procedure on Shoot Tips Regeneration and Organic Acid Content

#### 2.1.1. Regeneration

A DV cryopreservation protocol optimized for *P. yatabeanus* axillary shoot tips in the previous studies [10,11,12] was used, hereafter referred to as a “standard procedure”. Nodal sections cut from the donor plants (Figure 1A) were cultured for 3–4 days (Figure 1B); then, developed axillary shoot tips were dissected, and precultured in liquid medium with 10% sucrose for 31 h (Figure 1C), osmoprotected (OP), cryoprotected (CP) and cryopreserved (LN). After rewarming and unloading with a 35% sucrose solution for 40 min, shoot tips were subjected to a three-step regrowth starting with ammonium-free medium with 1 mg L^−1^ gibberellic acid (GA_3_) and 1 mg L^−1^ 6-benzylaminopurine (BA) (Figure 1D). Even with this optimized standard procedure, the regeneration percentage of shoot tips decreased by approximately 10% following the cryoprotection and cryopreservation treatments. The regeneration in fresh-control, PC, and PC-OP variants was almost 100%, while the regeneration of cryoprotected (LNC) and cryopreserved (LN) shoot tips was near 95% and 90%, respectively (data not presented). The development of cryoprotected (LNC) and cryopreserved (LN) shoot tips into plants was slower than in other treatments.

#### 2.1.2. Analysis of OAs in the Droplet-Vitrification Procedure

For the OA analysis in the standard DV procedure, shoot tips were sequentially precultured in one-component cryoprotective agent (CPA) (10% sucrose for 31 h (PC)), osmoprotected with a two-component CPA C4-35% (17.5% glycerol + 17.5% sucrose for 40 min (OP)), then cryoprotected with four-component CPA A3-80% (33.3% glycerol + 13.3% dimethyl sulfoxide + 13.3% ethylene glycol + 20.1% sucrose for 60 min on ice (CP)). After C/W in LN (LN), shoot tips were unloaded with a one-component CPA (35% sucrose for 40 min (UL)) and finally regrown with ammonium-free 3% sucrose medium for five days (LN-RM1).

The contents of 14 OAs and their normalized values calculated based on the corresponding OA level of untreated control shoot tips (fresh) are summarized in Table 1 and Table 2. The total contents of 14 OAs in control (fresh) shoot tips were 6150 ng·2 mg^−1^ fresh weight (FW) sample, and the normalized values of all OAs were set as 1.00. The total content of OAs gradually increased in the course of the cryopreservation procedure from fresh control to the PC (2.83-fold increase), osmoprotection (OP, 4.39-fold), and cryoprotection (CP, 5.42-fold), and peaked after cooling and rewarming (LN, 6.38-fold). After cryopreservation and five days of recovering on RM1 (ammonium-free) medium (LN-RM1), the total content of OAs dropped to 89% of the fresh control. These dynamics of OA total content highly correlated with the CPA concentration employed (*r* = 0.951).

The metabolic profiling revealed that the major five OAs, glycolic, malic, citric, malonic, and lactic, were the most abundant OAs in the fresh control shoot tips, representing 84.3% of the total OAs. Among those five major OAs, lactic acid and glycolic acid increased at LN stage by 37.5-fold and 10.2-fold, respectively. Malonic, malic, and citric acids remained similar to the fresh-control (0.8–1.4-fold increase).

The 14 OAs tested were categorized into three groups based on their dynamic pattern during the droplet-vitrification procedure. Group 1 was represented by the four OAs (2-hydroxybutyric, 3-hydroxypropionic, lactic, and glycolic acids) whose content highly increased during the cryopreservation procedure. On average, they showed a 10.9-fold increase at PC, a 42.2-fold increase at OP, a 62.9-fold increase at LNC, a 74.8-fold peak at LN, and remained 3.8-fold higher than the fresh control at the regrowth stage. The content of these four OAs accounted for 85.5% of the total OAs at the LN stage, while at fresh-control, they composed only 31.4% of the total OAs pool. Group 2, formed by pyruvic and 2-hydroxybutyric acids, demonstrated a 4.2-fold increase at LN stage and a slightly elevated (1.1–1.3-fold compared to control) content at the regrowth stage. In Group 3, eight OAs, including malonic, succinic, fumaric, α-ketoglutaric, malic, cis-aconitic, citric, and isocitric acids, showed little or no increase at LN stage (up to 1.3-fold) but decreased at the regrowth stage below the level of the fresh control (0.4–0.9 fold).

### 2.2. Effect of Additional Stress Imposed by Non-Optimum Conditions in the Droplet-Vitrification Procedure

#### 2.2.1. Regeneration

Variations from optimum conditions at different stages of the DV procedure were tested to explore their influence on shoot tip regeneration and OA content.

The treatments without preculture (noPC-RM1) or preculture with S-10% followed by 25% sucrose (S-25%) produced significantly lower regeneration after cryopreservation (60% and 62.2%, respectively) compared to the standard optimized procedure (91.1%) (Figure 2). A combination of no-preculture and ammonium-containing regrowth medium (noPC-RM2) resulted in an even lower post-LN regeneration of 40.6%. Cryopreservation without osmoprotection (noOP) or cooling and rewarming using a vial instead of aluminum foils (Vial) showed the most inferior post-LN regeneration (30.0% and 28.2%, respectively). These results indicate that *P. yatabeanus* shoot tips are sensitive to osmotic stress induced by high (25%) sucrose concentrations during the preculture stage or by a vitrification solution applied without preculture or osmoprotection. Shoot tips cryopreserved using vials instead of aluminum foils were possibly subjected to freezing injury caused by relatively slower C/W velocities. ROS-induced oxidative stress likely damaged cryopreserved shoot tips recovered on ammonium-containing regrowth medium (NoPC-RM2).

#### 2.2.2. Analysis of OAs under Non-Optimum Cryopreservation (Additional Stress) Conditions

A negative correlation (*r* = −0.375) was observed between the total OA content (Table 3 and Table 4) and regeneration of shoot tips cryopreserved using non-optimum conditions (Figure 2). In general, the effect of cryopreservation treatment modification on OA content was less prominent than that of cryopreservation procedure stages on fresh shoot tips (Table 1 and Table 2). It is also interesting that different treatments caused different responses regarding individual OA levels. In detail, the total OA content in shoot tips cryopreserved without preculture (noPC) or osmoprotection (noOP) was slightly higher than under standard protocol conditions. This tendency was most prominent in Groups 1 and 2 OAs. Total OAs content in the variant with high sucrose preculture (S-25%) was lower than that under standard conditions. Still, a minor increase in the content of malonic, malic, citric, and isocitric acids was observed. Cooling-rewarming using cryovials (Vial) caused elevation of lactic, malonic, succinic, fumaric, 2-hydroxyglutaric, and α-ketoglutaric acids but had little effect on the total OAs value due to a decrease in other OAs content (Table 3 and Table 4). In the procedure without preculture, the content of most OAs decreased after 5 days of recovering on an ammonium-free or ammonium-containing medium (treatments noPC-RM1 and noPC-RM2).

### 2.3. Effect of Plant Vitrification Solution

#### 2.3.1. Regeneration

Among the PVSs tested, A3-80% produced the highest regeneration after cryopreservation (90.6%) compared to other VSs (67.5~70.9%) (Figure 3). A four-component PVS2 and two-component B5-85% showed similar survival and regeneration in cryoprotected control (LNC) shoot tips but lower regeneration of cryopreserved shoot tips compared to A3-80% treatment, indicating a likely freezing injury. In the A3-90% treatment, recovery of both cryoprotected and cryopreserved shoot tips was lower than those in the A3-80% treatment, implying a high level of cytotoxicity imposed by highly concentrated PVS.

#### 2.3.2. Analysis of OAs in Different Plant Vitrification Solution Treatments

The vitrification solutions tested had a minor impact on the total concentration of OAs in cryopreserved shoot tips (Table 5 and Table 6). Treatments with highly concentrated four-component A3-90% and a two-component B5-85% induced a slightly higher total OA content than the standard protocol employing A3-80%. This increment was primarily due to lactic, 2-hydroxybutyric, pyruvic, 2-hydroxyglutaric and succinic acids. At the same time, A3-90% treatment caused a minor reduction of six out of eight OAs from Group 3. In general, modification of the PVS composition induced minor changes in the content of individual and total OAs.

### 2.4. Effect of Regrowth Conditions

#### 2.4.1. Regeneration

Following the standard DV procedure, cryopreserved shoot tips were regrown using six variants of the three-step regrowth process. The standard regrowth treatment included step 1 on RM1 (ammonium-free MS medium + GA_3_ 1 mg L^−1^ + BA 1 mg L^−1^) for five days, step 2 on RM2 (standard MS medium + GA_3_ 1 mg L^−1^ + BA 1 mg L^−1^) for 23 days, and step 3 on MSF (MS medium without growth regulators) for 14 days.

The use of ammonium-free RM1 medium during the first 2 or 5 days after rewarming produced a significantly higher post-LN regeneration (85.4~93.3%) compared to ammonium-containing RM2 medium (61.7~63.5%) (Figure 4). Excluding growth regulators from the ammonium-free medium during the first recovery days resulted in the lowest regeneration (40.0%). These results confirm that, during the initial recovery steps, plant growth regulators GA_3_ and BA were the most critical requirement for the regeneration of cryopreserved shoot tips, followed by ammonium-free macroelements. The three-step regrowth process was adequate for successfully regenerating cryopreserved *P. yatabeanus* shoot tips.

#### 2.4.2. Analysis of OAs under Different Regrowth Conditions

In the standard cryopreservation protocol, the total OA content reached the maximum at the cryopreservation stage, then dropped down during the first two days of regrowth on RM1 medium (RM1(2d)), decreased further after five days (RM1(5d)) to a level of fresh-control and remained stable during the next 9 days of recovery on RM2 (RM1(5d)-RM2(9d)) (Table 7 and Table 8), implying drastic metabolic changes during C/W and the initial 5 days of recovery. This result coincides with the previously published data that 5 days of recovering on ammonium-free RM1 medium produced higher regeneration than the shorter ammonium-free periods of 3 or 1 days [11]. It is worth noting that the Group 1 OA contents (2-hydroxybutyric, 3-hydroxypropionic, lactic, and glycolic acid) remained elevated during the regrowth stage. The content of Group 2 OAs (pyruvic and 2-hydroxybutyric acids) remained similar to the fresh-control, while Group 3 OAs showed 0.4–0.9-fold lower levels during the recovery compared to control shoot tips. Using ammonium-containing medium during the first 5 days of recovery (RM2(5d)) produced a slightly lower content of succinic, fumaric, α-ketoglutaric, and *cis*-aconitic acids (Group 3) and an elevated concentration of all other measured OAs compared to standard ammonium-free RM medium (RM1(5d)). However, after 14 days of recovery, this difference between ammonium-free and ammonium-containing medium was mitigated, as evidenced by comparing treatments (RM2(5d)-RM2(9d)) and (RM1(5d)-RM2(9d)). Regrowth starting with the hormone-free medium (RM1HF(5d)-MSF(9d)) led to the highest content of Group 1 OAs among 14-day regrowth treatments, exceeding even those recorded after regrowth on ammonium-containing medium (Table 7 and Table 8).

## 3. Discussion

### 3.1. Regeneration in Droplet-Vitrification Procedure and Oxidative Stress

Using a systematic approach, we developed a DV protocol for *P. yatabeana* shoot tips. It contains the preparation of vigorous donor plantlets via applying a liquid overlay on the gelled medium, cryoprotection with alternative vitrification solution A3-80%, and three-step regrowth initially with ammonium-free medium [10,11,12]. In the series of experiments, regeneration of the cryopreserved shoot tips was improved from 20% to 91%, resulting in the optimized “standard” protocol. As a solution-based vitrification technique, the DV consists of pre-LN (preculture, osmoprotection, cryoprotection with a vitrification solution), LN (cooling and rewarming, unloading), and post-LN (regrowth) stages. In this study, we introduced variations from the standard (optimum) procedure at different protocol stages to understand better the negative impact of various factors and their possible correlation with metabolic changes indicated by the level of the 14 OAs. In general, stress or non-optimum conditions significantly impacted the regeneration of shoot tips after cryopreservation. Regeneration was reduced in descending order as follows: standard condition (90.8%) > modification of vitrification solution (PVS2, 68% or B5-85%, 67%) > preculture (S-25%, 62%; no-PC, 60%) > regrowth medium (ammonium-ion RM2, 64%; no growth regulators RM1HF-MSF, 40%) > no osmoprotection (no-OP, 30%), inappropriate container (Vial, 28%).

Low regeneration after cryopreservation in vials was likely caused by intracellular ice crystallization [2] due to insufficient cryoprotection. At the same time, shoot tips were vulnerable to osmotic stress and chemical toxicity induced by a vitrification solution [3]. A proper vitrification solution is crucial since it protects the explants from freezing injury while exposing them to osmotic stress and chemical cytotoxicity [21]. Regarding balancing the positive and negative impacts, alternative vitrification solution A3-80% was superior to other solutions tested. Even with the optimized procedure, plant shoot tips suffered stress and injuries during DV, i.e., excision, preculture, osmoprotection, cryoprotection, C/W, and unloading. All these stresses and injuries may drive the cells into a burst of reactive oxygen species (ROS) [22,23]. Oxidative stress-related genes and proteins were identified in plant materials during cryopreservation [24,25]. Gene expression and transcriptome analysis in *Arabidopsis* shoot tips cryoprotected with PVS2 and cryopreserved in LN also suggested that such treatment triggered the oxidative stress response [26,27]. ROS-induced oxidative stress usually reaches a maximum at the cryoprotection (vitrification solution), C/W, and unloading stages [4,6,23,28]. Antioxidants have been shown to improve post-cryogenic regeneration of various plant materials [5,22,23,29]. Although ammonium is a vital nitrogen source for plants, ammonium-ion may also inhibit recovery from freezing injury [30,31]. For example, omitting ammonium-ion from the regrowth medium in sweet potatoes improved regeneration by 61% [32]. Cryopreservation stress may reduce the metabolic activity of the shoot tips, and the key enzymes of ammonia nitrogen metabolism could be inactivated after rewarming, leading to the accumulation of toxic levels of ammonium [33], eventually leading to a burst of ROS [25] and initiating programmed cell death (PCD) [34]. Hence, omitting ammonium in the regrowth medium for 5–7 days after rewarming may benefit normal plant regeneration [11].

The post-transcriptional and metabolism level accompanies the adaptation of plants to oxidative stress [35]. Hence, it is interesting to track the effect of the cryopreservation procedure on metabolites closely associated with redox regulation and energy balance in plants, such as sugars and their derivatives, TCA cycle metabolites (organic acids), and amino acids [35]. Repairing from cryopreservation injuries is an energy-consuming process requiring large amounts of ATP; however, both ATP content and mitochondrial function were found to decrease significantly in fish and avian species sperm cells as well as in potato shoot tips following cryopreservation [13,36,37]. The metabolic rate in shoot tips of Australian plant species also reduced significantly [18]. Through the intensive study of the animal system, cryopreservation procedures were hypothesized to reduce mitochondrial function and metabolic rates, and cryopreservation survival was linked with the mitochondrial health of organisms [7,18]. Some amino acids, such as alanine, isoleucine, and leucine, decreased in cryopreserved turkey sperm. At the same time, glycine was increased, implying amino acids’ crucial role in the quality of post-thaw sperm [13].

### 3.2. Metabolomics Profiling of OAs in Cryopreservation

OAs are intermediates of carbon metabolism in plants, such as the TCA cycle, glycolysis, the glyoxylate cycle, etc., and are preferentially stored in the vacuole [38]. Many abiotic stresses, for example, drought, stimulate OA biosynthesis and their release from the roots [19,20]. OA-pathway genes are up- or down-regulated to fine-tune the adaptation to adverse conditions [19]. Foliar application of OAs (succinic acid) alleviated aluminum toxicity and increased alfalfa’s plant growth and root activity, likely due to the antioxidant system [39]. Among the OAs, malic, galacturonic, and succinic acid content significantly increased under long-term drought stress conditions, while methylmalonic, citric, and isocitric acid decreased [20].

In the DV cryopreservation procedure, the shoot tips were sequentially exposed to osmotic dehydration and cryoprotectant loading through bathing in cryoprotectant solutions with increasing concentrations, from 10% (preculture step) to 35% (osmoprotection step) and 80% (cryoprotection). After rewarming, the solution concentration decreased reversely from 80% (C/W) to 35% (unloading) and finally to 3% (regrowth). In the present study, metabolic profiling revealed that glycolic, malic, citric, malonic, and lactic acids were the most abundant OAs in the fresh control shoot tips of *P. yatabeana*. Malic and citric acids are the primary acids in most ripe fruits and vegetative plant parts [40]. With the increase in cryoprotectant content in the standard DV procedure, the total content of OAs gradually increased from the preculture (2.83-fold) to osmoprotection (4.39-fold) and cryoprotection (5.42). It peaked at cooling and rewarming (6.38-fold). After cryopreservation, the OA content dropped sharply at 2-day regrowth on RM1 medium (1.83-fold) and returned to the fresh-control level at day 5 of regrowth (0.89-fold) (Figure 5). This dynamic pattern of OAs was highly correlated with the total cryoprotectant concentration (*r* = 0.951), indirectly evidencing oxidative stress development and energy consumption during the DV procedure. A similar dynamic was observed in vitrification-based cryopreservation for malondialdehyde (MDA), an indicator of lipid peroxidation and oxidative stress [28] and caspase-3-like activity involved in the execution of programmed cell death (PCD) [41]. Total antioxidant concentration decreased reversely, particularly at cryoprotection and unloading steps [6]. It is acknowledged that by operating the TCA cycle and its side reactions, plants can regulate the tissues’ redox status and energy metabolism [40]. The TCA cycle may work in closed or open mode depending on the current metabolic needs, providing the metabolic pathways with OAs through malate and citrate valves. This often results in these two organic acids being among the most accumulated OAs in plant tissues [40], including fresh (control) *P. yatabeanus* shoot tips in this study. The content of malic and citric acids increased twice from the control level at the preculture step, followed by a gradual decrease and fell below the control level after 5 days of regrowth, possibly indicating the intensification of metabolism to support regeneration (Table 1). In addition, nearly all OAs of the main TCA reactions reduced concentration under various regrowth conditions (Table 7), suggesting their high consumption during regrowth.

The 14 OAs tested were grouped into three categories based on their dynamics during the DV procedure. In Group 1, the four highly increasing OAs (2-hydroxybutyric, 3-hydroxy propionic, lactic, and glycolic acids) raised 74.8-fold on average of the fresh control at the C/W stage. They remained elevated (3.8–8.1-fold) at the regrowth stage (Figure 5). Group 2 (pyruvic and 2-hydroxybutyric acids) increased 4.2-fold at the C/W stage but dropped to near fresh-control level or slightly higher (1.1–1.3-fold) at the regrowth stage. In Group 3, eight OAs (malonic, succinic, fumaric, α-ketoglutaric, malic, cis-aconitic, citric, and isocitric acid) were slightly increased (up to 1.3-fold) at the C/W stage but decreased below the control level (0.4–0.9-fold) during regrowth. This is in line with the general trend of the pool of TCA cycle metabolites such as fumaric, malic, isocitric, and succinic acid to decrease under oxidative stress due to the disturbance with TCA cycle-related energy metabolism in mitochondria [35].

This study firstly aimed to monitor the profiles of OAs during the DV procedure to find a possible metabolic signature of oxidative stress and energy provision change and, secondly, to provide comprehensive insights by comparison of the metabolites (OAs) contents with post-cryopreservation regeneration of shoot tips using various treatment sets: non-optimized cryopreservation protocol, the varying composition of vitrification solution and regrowth conditions. Though a high correlation coefficient (*r* = 0.951) was detected between the cryoprotectant concentration and total OA content in the DV procedure, the relationship between regeneration and OA content was not straightforward (*r* = −0.321~−0.559). With the Group 1 OAs, the negative correlation coefficients of −0.508 (non-optimum protocol), −0.539 (vitrification solution), and −0.521 (regrowth conditions) were noticed between the LN regeneration and four OA contents. All the stresses during the DV procedure are known to trigger ROS-induced oxidative stress. The four OAs, i.e., 2-hydroxybutyric, 3-hydroxy propionic, lactic, and glycolic acids, synthesized by the shoot tips may help cope with cryoprotectant-induced osmotic stress and chemical cytotoxicity [23].

In this study, lactic and glycolic acid were among the five major OAs highly biosynthesized (Group 1) during the DV procedure. They were the most abundant among OAs tested, accounting for 81% of the total OAs content at C/W steps. Lactic acid bacteria (lactobacilli), such as *Streptococcus salivarius*, notably have antioxidant systems for stabilizing free radical levels and gained interest as probiotics in the health industry [42]. Lactic acid is an intermediate of anaerobic glycolysis, reflecting the reduced oxygen supply during the DV procedure. Glycolic acid is biosynthesized through the glyoxylate cycle, which uses lipids as a respiration substrate in plants and microbes [43].

Interestingly, glycolic acid was accumulated in high amounts even in fresh (control) shoot tips of *P. yanabeanus*, which was probably the specifics of this species. Poly (lactic-co-glycolic acid) (PLGA), a biodegradable functional polymer made from lactic acid and glycolic acid polymerization, is widely used in pharmaceuticals and medical engineering materials as a drug carrier [44,45]. In mammalian models, 2-hydroxybutyric acid is primarily produced during L-threonine metabolism or glutathione synthesis. It may be elevated by oxidative stress or the detoxification of exogenous substances in the liver and thus enriched in COVID-19 patients [43]. It is mainly promoted in the metabolic pathways of central carbon metabolism in cancer, protein digestion, mineral absorption, prostate cancer, and severe preeclampsia [46]. It drastically increased from fresh control (1.2 ng·2 mg^−1^ FW sample) to the LN step (256.9 ng·2 mg^−1^ FW sample, 209.3-fold) in the standard protocol condition. In the osmotic shock condition without preculture (NoPC-LN) or osmoprotection (NoOP-LN), they jumped to 372.8- and 407.5-fold. 3-Hydroxypropionic acid as root exudates improves nutrients, such as phosphorus, use efficiency [47]. Further intensive studies are needed to investigate the possibility of oxidative stress-related biomarkers in the cryopreservation of plants.

## 4. Materials and Methods

### 4.1. Plant Material, In Vitro Propagation and Preparation of Mother Plants

About 30 seeds of *Pogostemon yatabeanus* germinated in vitro after the dormancy breaking (hot water treatment followed by cold stratification), as described previously [10,11,12].

In vitro-developed plants were propagated via repeated subcultures using single node segments with hormone-free Murashige and Skoog medium (MSF, [48]) with 30 g L^−1^ sucrose, 3.0 g L^−1^ gellan gum (MB Cell, Seoul, Korea) in culture vessels (90 mm × 40 mm). For vigorous growth, liquid MSF medium (15 mL/vessel) was overlayed on top of gellan gum-gelled medium 10 days after inoculation. For subculture, six nodal segments per vessel were inoculated, and the cultures were kept at 25 °C under a 16/8 h light/dark photoperiod, light provided by one fluorescent lamp (40 μE m^−2^ s^−1^).

### 4.2. Experimental Design of Treatments in the Droplet-Vitrification Procedure

#### 4.2.1. Standard Droplet-Vitrification Procedure

After repeated standard subcultures for six weeks, shoot tips, 1.5 mm long with 1–2 lateral leaves, were excised from 3–4-day-old single nodal sections. These shoot tips were used in the experiments described below, ten to 13 shoot tips per experimental condition, and the experiments were replicated 3–4 times (*n* = 30–52).

In the standard cryopreservation procedure, shoot tips were precultured in 10% sucrose (S-10%) for 31 h, osmoprotected with C4-35% (17.5% glycerol + 17.5% sucrose) for 40 min, then cryoprotected with a vitrification solution A3-80% (33.3% glycerol + 13.3% dimethyl sulfoxide + 13.3% ethylene glycol + 20.1% sucrose) for 60 min on ice. Then, shoot tips were placed in drops of ice-cold A3-80% on aluminum foil strips (7 × 20 mm × 50 µm) and plunged directly in LN for a minimum of 1 h. For rewarming, foil strips with shoot tips were transferred to 20 mL pre-heated (40 °C) 35% sucrose (S-35%) solution and kept for 40 min. The explants retrieved from S-35% were blotted dry on filter paper and transferred to regrowth medium 1 (RM1) composed of ammonium–free MS medium with 1 mg L^−1^ gibberellic acid (GA_3_), 1 mg L^−1^ benzyl adenine (BA), 30 g L^−1^ sucrose, 3.0 g L^−1^ gellan gum, and cultured in the dark for 5 days. Next, explants were transferred to RM2 medium (the same as RM1 except with a standard concentration of NH_4_NO_3_) and cultured under one florescent lamp (40 µE m^−2^ s^−1^) for 23 days. The developed shoots were transferred to a hormone-free MS medium (MSF) for 2 weeks.

#### 4.2.2. Sets of Experimental Conditions

Variations in the cryopreservation procedure steps, including stress conditions, plant vitrification solution composition, and regrowth conditions, were tested during the DV procedure (Table 9). During the experiment of each factor, other conditions remained the same as in the standard DV procedure (indicated as “standard” in Table 9).

### 4.3. Analysis of OAs

#### 4.3.1. Chemicals and Reagents

The 14 OA standards (pyruvic, lactic, glycolic, 2-hydroxybutyric, 3-hydroxybutyric, malonic, succinic, fumaric, α-ketoglutaric, malic, 2-hydroxyglutaric, *cis*-aconitic, citric, and isocitric acid), and *O*-methoxyamine hydrochloride were purchased from Sigma-Aldrich (St. Louis, MO, USA). *N*-Methyl-*N*-(tert-butyldimethylsilyl) trifluoroacetamide (MTBSTFA) + 1% *tert*-butyldimetheylchlorosilane (TBDMCS) was obtained from Thermo Scientific (Bellefonte, PA, USA). Toluene, diethyl ether, ethyl acetate, and sodium chloride of pesticide grade were supplied by Kanto Chemical (Tokyo, Japan). All other chemicals were of analytical grade and used as received.

#### 4.3.2. Sample Preparation for Profiling Analysis of OAs by GC-MS/MS

The analyses of OAs were performed in two independent biological experiments. In the first round experiment, 20 shoot tips per condition were used in 29 conditions. For the second round experiment, 50 shoot tips per condition were excised from the nodal sections and used in the experiments (Table 9). After the treatments, shoot tips (55.9–246 mg FW) were briefly wiped with Kimtech wipers (Kimberly-Clark, Irving, TX, USA).

Profiling analysis of OAs in *P. yatabeanus* shoot tips was performed using methoximation-*tert* butyldimethylsilyl (TBDMS) derivatives, according to published methods [49,50,51]. Briefly, *P. yatabeanus* shoot tips were homogenized by adding distilled water for a 20 mg/mL concentration. The mixture was centrifuged and filtered through the Whatman qualitative filter (Sigma-Aldrich, St. Louis). Then, dilute 100 μL sample (2 mg) to 1 mL of distilled water containing 0.1 μg of 3,4-dimethoxybenzoic acid as an internal standard. *O*-Methoxyamine hydrochloride (1 mg) was spiked in the aqueous phase, adjusted to pH ≥ 12 using 5.0 M sodium hydroxide, and then reacted at 60 °C for 60 min. After the aqueous phase was adjusted to pH ≤ 2 with 10% sulfuric acid, saturated with sodium chloride. Samples were extracted with diethyl ether (3.0 mL) and ethyl acetate (2.0 mL). Organic solvents were evaporated to dryness under an N_2_ stream at 40 °C. Toluene (10 μL) and MTBSTFA + 1% TBDMCS (20 μL) derivatives were spiked in the dry residues and then transferred to auto vials for gas chromatography–tandem mass spectrometry (GC-MS/MS) analysis.

#### 4.3.3. Analytical Conditions of GC-MS/MS Analysis

The analyses of OAs were performed using a Shimadzu 2010 plus gas chromatograph coupled with a Shimadzu TQ 8040 triple quadruple mass spectrometer (Shimadzu, Kyoto, Japan). The column used was an Ultra-2 (5% phenyl–95% methylpolysiloxane bonded phase; 25 m × 0.20 mm i.d., 0.11 µm film thickness) cross-linked capillary column (Agilent Technologies, Atlanta, GA, USA). Samples (1 μL) were injected in split-injection mode (10:1 and 100:1). The temperatures of the injector, interface, and ion sources were 230, 260, and 300 °C, respectively. Helium (0.5 mL min^−1^) and argon were used as a carrier and collision gases, respectively. Ionization was performed in electron impact ionization mode at 70 eV. The collision energy ranged from 3 to 45 V in increments of 3 V in multiple reaction monitoring modes. Shimadzu’s data acquisition software was used for quantifying organic acids in samples and systematic data analysis.

#### 4.3.4. Normalized Pattern Analysis

Levels of OAs were calculated using appropriate calibration curves. The levels of OAs were normalized to the corresponding values of each metabolite in the fresh (untreated) shoot tips or a standard DV procedure. Normalizations were evaluated using Microsoft Excel ver. 2019 (Microsoft, Redmond, WA, USA) [52].

### 4.4. Recovery Assessment and Statistical Analysis

Survival was evaluated as the percentage of shoot tips showing the appearance of green tissues 14 days after rewarming. Regeneration was calculated six weeks after rewarming as the percentage of shoot tips that developed into healthy-looking plantlets over 8 mm in height with leaves and roots. Ten to 13 shoot tips were used per experimental condition, and the experiments were replicated 3–4 times.

The analyses of OAs were performed in two independent biological experiments using 20 and 50 shoot tips per condition, respectively, and similar patterns were observed for individual OAs in response to the change in experimental conditions. But for the benefit of consistency, the data of the second experiment were presented. Regeneration data from all experiments were analyzed by analysis of variance (ANOVA) and Duncan’s multiple range test (*p* < 0.05) using SAS on Demand for Academics software (SAS Institute Inc., Cary, NC, USA). Results are presented as percentages with standard deviations. The correlation coefficient (CORREL, *r*) was calculated by comparing total OA contents and CPA concentration of each stage or LN regeneration with the corresponding conditions using Microsoft Excel.

## 5. Conclusions

Cryopreservation of vegetative plant germplasm is a multi-step process, and every step imposes additional stress. This includes osmotic dehydration, chemical toxicity from the concentrated CPA solutions, increased cytoplasm viscosity, and oxidative stress. These stress conditions are associated with the extensive shift in cell metabolism, which is poorly understood. In the present work, we investigated the effect of the individual steps of the cryopreservation protocol and their modifications on the content of 14 organic acids associated with the TCA cycle and its secondary reactions in the shoot tips of the endangered Korean species *P. yatabeanus*. In the optimized cryopreservation procedure, the total OA content correlated with the concentration of the cryoprotectant solutions: it increased during CPA treatment until reaching the maximum at the C/W step, then dropped down during regrowth.

Among OAs, glycolic, malic, citric, malonic, and lactic acids were the most abundant in control (untreated) shoot tips. The most profound effect of the cryopreservation procedure was recorded for 2-hydroxybutyric, 3-hydroxy propionic, lactic, and glycolic acids, which raised 74.8-fold on average at the cooling–rewarming step compared to untreated control. Variations from the optimum protocol conditions affected individual OAs’ content differently. For example, the levels of lactic, 2-hydroxybutyric, pyruvic, and 2-hydroxyglutaric acids were elevated in response to osmotic stress caused by vitrification solution applied without preculture or osmoprotection. In contrast, step-wise preculture with 10% and 25% sucrose led to a minor increase in malonic, malic, citric, and isocitric acids and a reduction in the other OAs tested. Cooling–rewarming using vials instead of foil strips induced higher lactic, 2-hydroxyglutaric, malonic, succinic, fumaric, and α-ketoglutaric acids accumulation. There was no correlation between OA content and regeneration of shoot tips after different treatments. However, most OAs of the main TCA cycle reactions were reduced during regrowth.

These data suggest that OAs are involved in stress response and adaptation of shoot tips to adverse conditions of the cryopreservation process, possibly as a part of the system regulating oxidative stress development and providing energy and substrates for intensified metabolic reactions during regrowth. This study provides a first glance at the OAs metabolism under cryopreservation treatment. Still, we hope it would encourage other researchers to look at the metabolomics of plant materials, which may assist in protocol optimization for the endangered species that are often difficult to cryopreserve.

## Figures and Tables

**Figure 1 plants-12-03489-f001:**
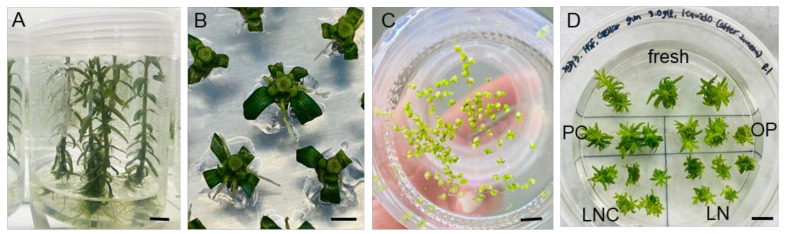
(**A**) Overlay of liquid hormone-free medium on top of gellan gum-gelled medium produced vigorously grown *P. yatabeanus* donor plantlets after 6 weeks. (**B**) Nodal sections from the donor plantlets were cultured for 3–4 days to induce the axillary shoot tips. (**C**) Shoot tips were excised and precultured in hormone-free MS liquid medium with 10% sucrose (S-10%) for 31 h. (**D**) Shoot tips after each step of the DV procedure were collected and subjected to regrowth using RM1 medium (ammonium-free, GA_3_+BA) followed by RM2 (ammonium-containing, GA_3_+BA) for 4 weeks in total. Fresh—excised shoot tips without treatment; PC—preculture (S-10%, 31 h); OP—preculture and osmoprotection (C4-35%, 40 min); LNC—preculture, osmoprotection and cryoprotection (A3-80% ice, 60 min) but no cooling–rewarming; LN—all steps including cryopreservation (cooling/rewarming using aluminum foil strips). Scale bars A and D, 10 mm; B and C, 2 mm.

**Figure 2 plants-12-03489-f002:**
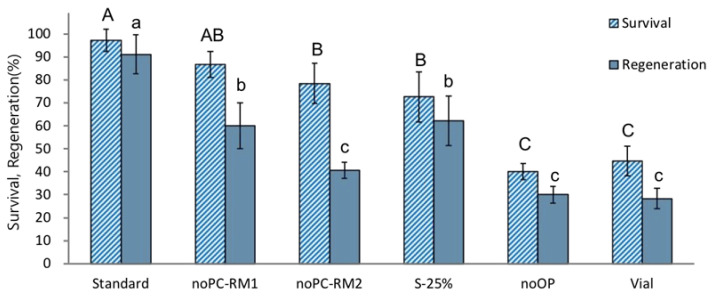
Effect of additional stress (non-optimum conditions) at different stages of the droplet-vitrification procedure on survival and regeneration of cryopreserved *P. yatabeanus* shoot tips. Standard, standard cryopreservation protocol conditions: preculture with 10% sucrose for 31 h, osmoprotection with C4-35% for 40 min, cryoprotection with vitrification solution A3-80% for 60 min on ice, freezing in LN and rewarming using aluminum foil strips, unloading with 35% sucrose for 40 min, regrowth on ammonium-free MS medium + GA_3_ 1 mg L^−1^ + BA 0.5 mg L^−1^ (RM1) for five days followed by ammonium-containing MS medium + GA_3_ 1 mg L^−1^ + BA 0.5 mg L^−1^ (RM2) for 23 days and ammonium-containing hormone-free MS for 14 days.; NoPC, protocol excluding preculture; NoPC-RM1, preculture excluded, regrowth on ammonium-free medium for the first 5 days; NoPC-RM2, preculture excluded, regrowth on ammonium-containing medium for the first 5 days; S-25%, preculture modified (10% sucrose for 31 h followed by 25% sucrose for 17 h); NoOP, osmoprotection excluded; Vial, freezing/rewarming in cryovials instead of aluminum foil strips. Values followed by the same letter are not significantly different between treatments at *p* < 0.0 (Duncan’s Multiple Range test).

**Figure 3 plants-12-03489-f003:**
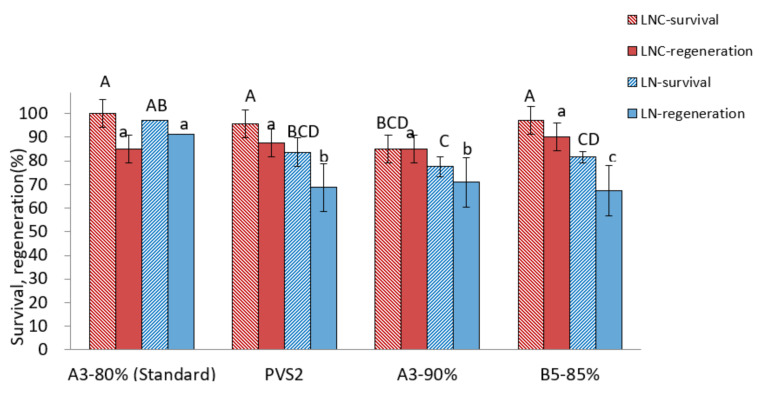
Effect of vitrification solutions on survival and regeneration of cryoprotected control (LNC) and cryopreserved (LN) *P. yatabeanus* shoot tips. Values followed by the same letter are not significantly different between treatments at *p* < 0.0 (Duncan’s Multiple Range test).

**Figure 4 plants-12-03489-f004:**
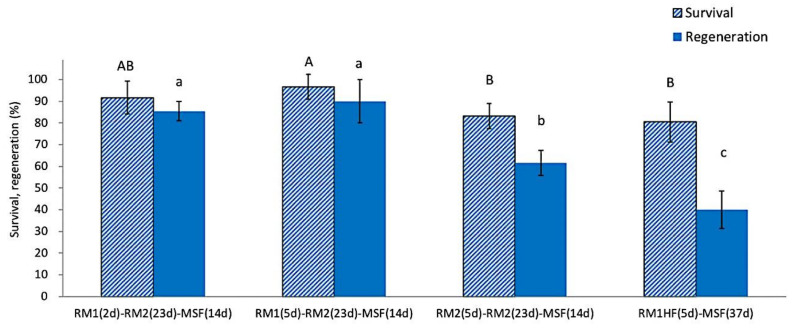
Effect of three-step regrowth using different media on survival and regeneration of cryopreserved *P. yatabeanus* shoot tips. RM1, ammonium-free MS medium + GA_3_ 1 mg L^−1^ + BA 0.5 mg L^−1^; RM2, ammonium-containing MS medium with the same growth regulators; RM1(HF), ammonium-free MS medium without growth regulators; d, days. Treatment consisting of RM1 (5 d)–RM2 (23 d)–MSF (14 d) corresponds to regrowth conditions in the standard protocol. Values followed by the same letter are not significantly different between treatments at *p* < 0.0 (Duncan’s Multiple Range test).

**Figure 5 plants-12-03489-f005:**
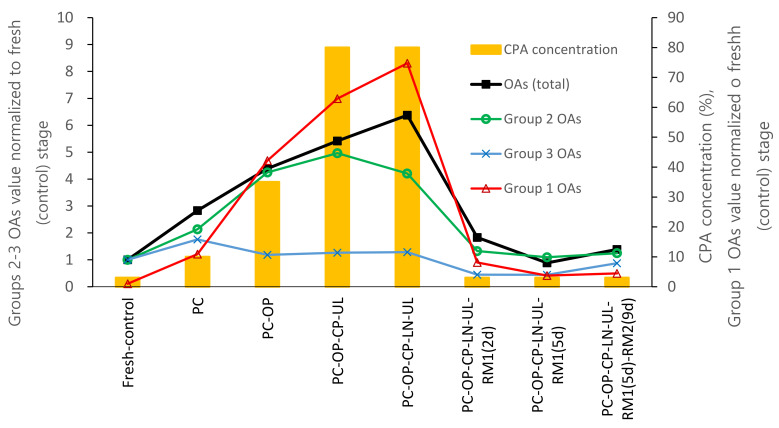
Dynamics of three groups of OAs in *P. yatabeanus* shoot tips during droplet-vitrification procedure relative to a total concentration of cryoprotectants (CPA). Fresh-control, excised shoot tips without treatment; PC, preculture; OP, osmoprotection; CP, cryoprotection (vitrification solution); UL, unloading; LN, cooling–rewarming in liquid nitrogen; RM1 and RM2, respectively, ammonium-free and ammonium-containing regrowth medium; d, days.

**Table 1 plants-12-03489-t001:** The content of 14 OAs in *P. yatabeanus* shoot tips following different stages of the droplet-vitrification procedure.

No.		Organic Acid	Amount (ng·2 mg^−1^ FW Sample)					
			Fresh *	PC	OP	LNC	LN	LN-RM1
1	Group 1	Lactic acid	459.3	3761.4	13,322.7	14,250.9	17,204.5	1788.5
2		Glycolic acid	1437.6	5629.3	8784.0	12,258.2	14,584.5	2069.7
3		2-Hydroxybutyric acid	1.2	26.9	144.0	221.2	256.9	9.0
4		3-Hydroxypropionic acid	35.6	337.8	585.6	1135.3	1497.7	84.0
5	Group 2	Pyruvic acid	30.9	43.9	96.2	89.3	105.4	37.4
6		2-Hydroxyglutaric acid	152.7	434.3	822.6	1075.2	764.4	149.0
7	Group 3	Malonic acid	864.2	695.0	864.3	513.0	715.5	128.3
8		Succinic acid	293.6	555.5	388.2	461.7	394.8	184.6
9		Fumaric acid	77.6	151.6	257.6	216.4	187.2	46.3
10		α-Ketoglutaric acid	136.1	117.7	130.9	96.0	76.6	77.7
11		Malic acid	1229.1	2613.8	791.0	1406.1	1436.9	356.2
12		*cis*-Aconitic acid	105.1	161.9	80.3	83.9	91.2	47.3
13		Citric acid	1194.4	2516.8	623.5	1329.4	1673.5	395.1
14		Isocitric acid	133.4	370.2	122.8	187.7	223.1	70.8
		Total OAs	6150.6	17,416.1	27,013.6	33,324.3	39,212.3	5443.9

* Stages of the cryopreservation procedure: fresh, untreated control; PC, preculture only; OP, preculture and osmoprotection (PC-OP); LNC, preculture, osmoprotection, cryoprotection and unloading (PC-OP-CP-UL); LN, preculture, osmoprotection, cryoprotection, cooling–rewarming and unloading (PC-OP-CP-LN-UL); LN-RM1, preculture, osmoprotection, cryoprotection, cooling–rewarming, unloading and recovery for 5 days on RM1 medium [PC-OP-CP-LN-UL-RM1(5d)]. Conditions of the cryopreservation protocol stages: PC, preculture with 10% sucrose for 31 h; OP, osmoprotection with C4-35% for 40 min; CP, cryoprotection with vitrification solution A3-80% for 60 min on ice; LN, cooling in liquid nitrogen using aluminum foil strips; UL, unloading with 35% sucrose for 40 min; RM1, regrowth with ammonium-free MS medium + GA_3_ 1 mg L^−1^ + BA 0.5 mg L^−1^ for five days. FW, fresh weight.

**Table 2 plants-12-03489-t002:** Heat map of the normalized values of 14 OAs in *P. yatabeanus* shoot tips following different stages of the droplet-vitrification procedure.

No.		Organic Acid	Normalized Value *					
			Fresh **	PC	OP	LNC	LN	LN-RM1
1	Group 1	Lactic acid	1	8.19	29.01	31.03	37.46	3.89
2		Glycolic acid	1	3.92	6.11	8.53	10.15	1.44
3		2-Hydroxybutyric acid	1	21.92	117.3	180.22	209.27	7.32
4		3-Hydroxypropionic acid	1	9.5	16.47	31.93	42.13	2.36
5	Group 2	Pyruvic acid	1	1.42	3.11	2.89	3.41	1.21
6		2-Hydroxyglutaric acid	1	2.84	5.39	7.04	5	0.98
7	Group 3	Malonic acid	1	0.8	1	0.59	0.83	0.15
8		Succinic acid	1	1.89	1.32	1.57	1.34	0.63
9		Fumaric acid	1	1.95	3.32	2.79	2.41	0.6
10		α-Ketoglutaric acid	1	0.86	0.96	0.71	0.56	0.57
11		Malic acid	1	2.13	0.64	1.14	1.17	0.29
12		*cis*-Aconitic acid	1	1.54	0.76	0.8	0.87	0.45
13		Citric acid	1	2.11	0.52	1.11	1.4	0.33
14		Isocitric acid	1	2.78	0.92	1.41	1.67	0.53
		Total OAs	1.00	2.83	4.39	5.42	6.38	0.89

* Values normalized to the corresponding value of each OA in the fresh-control. ** Treatment codes correspond to those in Table 1. Gradations of blue color indicate the reduction of OA value from the fresh control level. Gradations of yellow-to-red color indicate the increased OA level compared to fresh control. Color depth corresponds to the degree of difference from the control level.

**Table 3 plants-12-03489-t003:** The levels of 14 OAs in *P. yatabeanus* shoot tips cryopreserved using non-optimum conditions (additional stress effect).

No.		Organic Acid	Amount (ng·2 mg^−1^ FW Sample) *						
			Standard **	noPC	S-25%	noOP	Vial	NoPC-RM1	NoPC-RM2
1	Group 1	Lactic acid	14,952.6	20,397.4	9209.1	23,347.7	18,538.6	2295.1	2823.4
2		Glycolic acid	14,150.5	13,035.3	11,541.5	13,038.8	10,130.9	4131.8	3688.8
3		2-Hydroxybutyric acid	309.1	457.6	160.2	500.2	314.0	16.8	15.7
4		3-Hydroxypropionic acid	2051.3	1762.7	1469.9	1797.6	1025.3	254.1	199.4
5	Group 2	Pyruvic acid	112.2	143.4	83.2	150.4	93.6	56.5	47.5
6		2-Hydroxyglutaric acid	586.3	1031.3	477.7	1116.6	793.3	317.7	272.3
7	Group 3	Malonic acid	337.3	81.4	450.4	69.6	480.2	201.3	94.2
8		Succinic acid	279.7	211.9	265.5	200.8	335.8	104.1	42.8
9		Fumaric acid	101.3	41.5	45.4	43.0	124.0	30.2	24.8
10		α-Ketoglutaric acid	68.1	68.3	66.0	65.2	85.4	61.6	54.7
11		Malic acid	1700.7	399.9	1961.0	492.4	473.0	1060.5	467.4
12		*cis*-Aconitic acid	75.4	35.9	40.6	29.5	42.0	27.2	19.2
13		Citric acid	1884.8	650.0	2296.8	559.1	315.7	957.1	304.2
14		Isocitric acid	189.9	109.1	241.2	107.9	73.9	338.8	49.0
		Total OAs	36,799.2	38,425.6	28,308.3	41,518.8	32,825.8	9852.8	8103.5

* OA content was measured immediately after unloading (treatments Standard, noPC, S-25%, noOP and Vial) or after 5 days of regrowth on RM1 or RM2 medium (treatments NoPC-RM1 and NoPC-RM2). ** Variations of the cryopreservation protocol. Standard, standard cryopreservation procedure: preculture with 10% sucrose for 31 h, osmoprotection with C4-35% for 40 min, cryoprotection with vitrification solution A3-80% for 60 min on ice, freezing in LN and rewaming using Al foil strips, unloading with 35% sucrose for 40 min, regrowth with ammonium-free MS medium + GA3 1 mg L^−1^ + BA 0.5 mg L^−1^ (RM1) for five days. NoPC, protocol excluding preculture; NoPC-RM1, preculture excluded, regrowth on ammonium-free medium for the first 5 days; NoPC-RM2, preculture excluded, regrowth on ammonium-containing medium for the first 5 days; S-25%, preculture modified (10% sucrose for 31 h followed by 25% sucrose for 17 h); NoOP, osmoprotection excluded; Vial, cooling/rewarming in cryovials instead of aluminum foil strips. FW, fresh weight.

**Table 4 plants-12-03489-t004:** Heat map of the normalized values of 14 OAs in *P. yatabeanus* shoot tips cryopreserved using non-optimum conditions (additional stress effect).

No.		Organic Acid	Normalized Value *						
			Standard	noPC	NoPC-RM1	NoPC RM2	S-25%	noOP	Vial
1	Group 1	Lactic acid	1.0	1.36	0.15	0.19	0.62	1.56	1.24
2		Glycolic acid	1.0	0.92	0.29	0.26	0.82	0.92	0.72
3		2-Hydroxybutyric acid	1.0	1.48	0.05	0.05	0.52	1.62	1.02
4		3-Hydroxypropionic acid	1.0	0.86	0.12	0.10	0.72	0.88	0.50
5	Group 2	Pyruvic acid	1.0	1.28	0.50	0.42	0.74	1.34	0.83
6		2-Hydroxyglutaric acid	1.0	1.76	0.54	0.46	0.81	1.90	1.35
7	Group 3	Malonic acid	1.0	0.24	0.60	0.28	1.34	0.21	1.42
8		Succinic acid	1.0	0.76	0.37	0.15	0.95	0.72	1.20
9		Fumaric acid	1.0	0.41	0.30	0.25	0.45	0.42	1.22
10		α-Ketoglutaric acid	1.0	1.00	0.91	0.80	0.97	0.96	1.25
11		Malic acid	1.0	0.24	0.62	0.27	1.15	0.29	0.28
12		*cis*-Aconitic acid	1.0	0.48	0.36	0.25	0.54	0.39	0.56
13		Citric acid	1.0	0.34	0.51	0.16	1.22	0.30	0.17
14		Isocitric acid	1.0	0.57	1.78	0.26	1.27	0.57	0.39
		Total OAs	1.0	1.04	0.27	0.22	0.77	1.13	0.89

* Values normalized to the corresponding value of each OA in the Standard procedure. Treatment codes correspond to those in Table 3. Gradations of blue color indicate the reduction of OAs value from their levels in standard procedure. Gradations of red color indicate the increased OAs level compared to the standard procedure. Color depth corresponds to the degree of difference.

**Table 5 plants-12-03489-t005:** The levels of 14 OAs in *P. yatabeanus* shoot tips cryopreserved using different plant vitrification solutions.

No.		Organic Acid	Amount (ng·2 mg^−1^ FW Sample)			
			A3-80% * (Standard)	PVS2	A3-90%	B5-85%
1	Group 1	Lactic acid	14,952.6	16,140.0	17,992.4	17,154.3
2		Glycolic acid	14,150.5	14,255.9	13,459.0	12,180.4
3		2-Hydroxybutyric acid	309.1	454.8	388.6	335.6
4		3-Hydroxypropionic acid	2051.3	2147.4	1944.3	1477.2
5	Group 2	Pyruvic acid	112.2	114.0	138.7	121.7
6		2-Hydroxyglutaric acid	586.3	886.5	901.5	1065.8
7	Group 3	Malonic acid	337.3	288.4	316.0	363.9
8		Succinic acid	279.7	302.3	340.9	438.1
9		Fumaric acid	101.3	39.2	45.4	46.3
10		α-Ketoglutaric acid	68.1	70.2	74.7	79.8
11		Malic acid	1700.7	1505.9	2336.4	2832.3
12		*cis*-Aconitic acid	75.4	43.5	49.7	39.2
13		Citric acid	1884.8	1415.5	1856.8	2875.3
14		Isocitric acid	189.9	167.9	245.4	328.4
		Total	36,799.2	36,916.4	39,708.6	39,234.7

* Different plant vitrification solutions used. FW, fresh weight.

**Table 6 plants-12-03489-t006:** The normalized levels of 14 OAs in *P. yatabeanus* shoot tips from the plant vitrification solutions test.

No.		Organic Acid	Normalized Value *			
			A3-80% ** (Standard)	PVS2	A3-90%	B5-85%
1	Group 1	Lactic acid	1.00	1.08	1.20	1.15
2		Glycolic acid	1.00	0.92	1.01	0.95
3		2-Hydroxybutyric acid	1.00	1.10	1.47	1.26
4		3-Hydroxypropionic acid	1.00	0.84	1.05	0.95
5	Group 2	Pyruvic acid	1.00	1.02	1.24	1.08
6		2-Hydroxyglutaric acid	1.00	1.38	1.51	1.54
7	Group 3	Malonic acid	1.00	0.64	0.86	0.94
8		Succinic acid	1.00	1.32	1.08	1.22
9		Fumaric acid	1.00	0.49	0.39	0.45
10		α-Ketoglutaric acid	1.00	1.11	1.03	1.10
11		Malic acid	1.00	1.14	0.89	1.37
12		*cis*-Aconitic acid	1.00	0.56	0.58	0.66
13		Citric acid	1.00	1.02	0.75	0.99
14		Isocitric acid	1.00	1.12	0.88	1.29
		Total	1.00	1.00	1.08	1.07

* Values normalized to the corresponding value of each OA in the Standard procedure (VS A3-80%-LN). ** Treatment codes correspond to those in Table 5. Gradations of blue color indicate the reduction of OAs values from their levels in standard procedure. Gradations of red color indicate the increased OAs level compared to the standard procedure. Color depth corresponds to the degree of difference.

**Table 7 plants-12-03489-t007:** The levels of OAs in *P. yatabeanus* shoot tips from the regrowth condition.

No.		Organic Acid	Amount (ng·2 mg^−1^ FW Sample)						
			Fresh	RM1 (2d)	RM1 (5d)	RM1(5d)- RM2(9d)	RM1HF(5d)- MSF(9d)	RM2 (5d)	RM2(5d)- RM2(9d)
1	Group 1	Lactic acid	459.3	3461.8	1788.5	1134.8	2725.4	2811.8	956.1
2		Glycolic acid	1437.6	5522.2	2069.7	2736.0	4739.7	3417.7	2159.3
3		2-Hydroxybutyric acid	1.2	16.8	9.0	10.7	21.4	13.0	7.2
4		3-Hydroxypropionic acid	35.6	257.1	84.0	173.0	218.2	149.1	88.8
5	Group 2	Pyruvic acid	30.9	38.0	37.4	39.4	44.8	47.3	47.9
6		2-Hydroxyglutaric acid	152.7	216.5	149.0	185.8	324.7	309.4	193.9
7	Group 3	Malonic acid	864.2	231.4	128.3	772.7	444.2	143.1	327.2
8		Succinic acid	293.6	100.2	184.6	122.4	65.8	76.2	103.1
9		Fumaric acid	77.6	38.2	46.3	45.8	34.0	32.4	33.7
10		α-Ketoglutaric acid	136.1	60.3	77.7	72.1	56.7	60.7	70.2
11		Malic acid	1229.1	558.6	356.2	1229.7	482.2	477.0	1198.7
12		*cis*-Aconitic acid	105.1	41.8	47.3	47.8	28.9	29.6	41.9
13		Citric acid	1194.4	628.9	395.1	1747.9	613.1	479.0	1409.9
14		Isocitric acid	133.4	88.8	70.8	221.3	66.6	76.3	189.0
		Total	6150.6	11,260.6	5443.9	8539.4	9865.8	8122.6	6826.9

Fresh, control (non-treated) shoot tips; RM1, ammonium-free MS medium + GA_3_ 1 mg L^−1^ + BA 0.5 mg L^−1^; RM2, ammonium-containing MS medium with the same growth regulators; RM1HF, ammonium-free MS medium without growth regulators; d, days.

**Table 8 plants-12-03489-t008:** The normalized levels of 14 OAs in cryopreserved *P. yatabeanus* shoot tips under different regrowth conditions.

No.		Organic Acid	Fresh	Normalized Value *					
				RM1 (2d)	RM1 (5d)	RM1(5d)- RM2(9d)	RM1HF(5d)- MSF(9d)	RM2 (5d)	RM2(5d)- RM2(9d)
1	Group 1	Lactic acid	1	7.54	3.89	3.89	5.93	6.12	2.08
2		Glycolic acid	1	3.84	1.44	1.44	3.30	2.38	1.50
3		2-Hydroxybutyric acid	1	13.69	7.33	7.33	17.43	10.59	5.87
4		3-Hydroxypropionic acid	1	7.23	2.36	2.36	6.14	4.19	2.50
5	Group 2	Pyruvic acid	1	1.23	1.21	1.21	1.45	1.53	1.55
6		2-Hydroxyglutaric acid	1	1.42	0.98	0.98	2.13	2.03	1.27
7	Group 3	Malonic acid	1	0.27	0.15	0.15	0.51	0.17	0.38
8		Succinic acid	1	0.34	0.63	0.63	0.22	0.26	0.35
9		Fumaric acid	1	0.49	0.60	0.60	0.44	0.42	0.43
10		α-Ketoglutaric acid	1	0.44	0.57	0.57	0.42	0.45	0.52
11		Malic acid	1	0.45	0.29	0.29	0.39	0.39	0.98
12		*cis*-Aconitic acid	1	0.40	0.45	0.45	0.28	0.28	0.40
13		Citric acid	1	0.53	0.33	0.33	0.51	0.40	1.18
14		Isocitric acid	1	0.67	0.53	0.53	0.50	0.57	1.42
		Total	1	1.83	0.89	1.13	1.60	1.32	1.25

* Values normalized to the corresponding value of each OA in the control (Fresh) shoot tips. RM1, ammonium-free MS medium + GA_3_ 1 mg L^−1^ + BA 0.5 mg L^−1^; RM2, ammonium-containing MS medium with the same growth regulators; RM1HF, ammonium-free MS medium without growth regulators; d, days. Gradations of blue color indicate the reduction of OA value from the fresh control level. Gradations of red color indicate the increased OA level compared to fresh control. Color depth corresponds to the degree of difference.

**Table 9 plants-12-03489-t009:** Set of experimental conditions during the droplet-vitrification procedure for LN regeneration and analyzing OA content during cryopreservation of *P. yatabeanus* shoot tips.

Factor Investigated	No.	Treatment Conditions	Code
Procedure stages	1	Fresh (untreated) control	fresh
	2	PC only	PC
	3	PC-OP only	PC-OP
	4	PC-OP-CP-UL	LNC
	5	PC-OP-CP-LN-UL	LN
	6	PC-OP-CP-LN-UL-RM1(5d)	LN-RM1(5d)
Additional stress (non-optimum protocol)	1	**No preculure**-OP-CP-LN-UL	NoPC
	2	**No preculure**-OP-CP-LN-UL-**RM1(5d)**	NoPC-RM1
	3	**No preculure**-OP-CP-LN-UL-**RM2(5d)**	NoPC-RM2
	4	PC-OP-CP-LN-UL	Standard *
	5	**PC(S-10%, 31 h→S-25%, 17 h)**-CP-LN-UL	S-25%
	6	PC-**no osmoprotection**-CP-LN-UL	NoOP
	7	PC-OP-CP-LN**(Vial)**-UL	Vial
Vitrification solution	1	PC-OP-CP**(PVS2 ice 60 m)**-UL	PVS2-LNC
	2	PC-OP-CP**(PVS2 ice 60 m)**-LN-UL	PVS2-LN
	3	PC-OP-CP**(A3-90% ice 60 m)**-LN-UL	A3-90%-LN
	4	PC-OP-CP**(A3-80% ice 60 m)**-UL	A3-80%-LNC (Standard)
	5	PC-OP-CP(A3-80% ice 60 m)-LN-UL	A3-80%-LN (Standard)
	6	PC-OP-CP**(B5-85% RT 60 m)**-LN-UL	B5-85%-LN
Regrowth steps and medium-type	1	PC-OP-CP-LN-UL-**RM1(2d)**	RM1(2d)
	2	PC-OP-CP-LN-UL-**RM1(5d)**	RM1(5d) (Standard)
	3	PC-OP-CP-LN-UL-**RM1(5d)-RM2(9d)**	RM1-RM2 (Standard)
	4	PC-OP-CP-LN-UL-**RM1(HF,5d)-MSF(9d)**	RM1(HF)-MSF
	5	PC-OP-CP-LN-UL-**RM2(5d)**	RM2(5d)
	6	PC-OP-CP-LN-UL-**RM2(5d)-RM2(9d)**	RM2-RM2

* Standard conditions: Nodal section-induced shoot tips (fresh) were subjected to a standard cryopreservation procedure which include preculture (PC, 10% sucrose, 31 h)—osmoprotection (OP, C4-35%, 40 min)—cryoprotection (CP, A3-80%, ice, 60 min)—cooling/warming using aluminum foil strips (LN)—unloading (UL, 35% sucrose, 40 min, solution changed after the first 15 min). Treatments in the table are the same as in the standard procedure if not stated otherwise. A3-80%, 33.3% glycerol + 13.3% dimethyl sulfoxide + 13.3% ethylene glycol + 20.1% sucrose; C4-35%, 17.5% glycerol + 17.5% sucrose; RM1, regrowth medium 1 (NH_4_NO_3_–free MS medium with 1 mg L^−1^ GA_3_ + 1 mg L^−1^ BA); RM1(HF), RM1 (NH_4_NO_3_–free MS medium) without growth regulators; RM2, regrowth medium 2 (ammonium containing MS medium with 1 mg L^−1^ GA_3_ + 1 mg L^−1^ BA); MSF, ammonium-containing MS medium without growth regulators. Standard regrowth conditions were set as RM1(5d)—RM2(23d)—MSF(14d). For each treatment, *n* = 30–52 shoot tips for LN regeneration, and *n* = 50 for OA analysis.

## Data Availability

Not applicable.

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
