# Peer review of "Dynamics of Organic Acids during the Droplet-Vitrification Cryopreservation Procedure Can Be a Signature of Oxidative Stress in Pogostemon yatabeanus"

_plants, 2023, doi:10.3390/plants12193489_

Round 1
Reviewer 1 Report
Metabolomic Dynamics of Organic Acids During the Droplet-Vitrification Procedure Can Be a Signature of Oxidative Stress 3 in Endangered Species Pogostemon yatabeanus
This manuscript takes a detailed look at optimising the cryopreservation protocol for P. yatabeanus using a range of treatments and optimisations for the droplet vitrification protocol, achieving a very high success rate with the optimised protocol. In addition, the authors analysed the organic acid content (14 different organic acids) to gain insight into their role during the different stages and treatments tested. Overall, this is an interesting area of research that can provide novel insights into cryopreservation that has not yet been assessed in plant, however there are some major issues with the experimental design with the organic acid analysis.
There is no mention of any replication, statistical analysis, or any reported standard errors for the organic acid analysis. Was no biological replicates or even technical replicates done for this analysis? The results of this study would be more relevant with proper replication and appropriate statistical analysis of the organic acid content rather than the large number of different treatments tested (over 20 different treatments only replicated once?).
Additional areas that need addressing are listed below:
Abbreviations of the cryopreservation treatments – there are a lot of different treatments tested and abbreviations used, this gets confusing as sometimes the treatment condition abbreviation or the code abbreviation from table 3 is used in the text and figures. Additionally, the abbreviations are not consistently applied (i.e. line 158 no-PC-LN-RM1 vs noPC-LN-RM1). Perhaps bolding the step where the different treatments is being applied would aid the reading in quickly identifying what treatment condition has changed for comparison. Or perhaps an overview figure highlighting the different treatments tested at each stage of the cryopreservation protocol.
Line 22 – “…to 28% with the cryovial” this sentence does not clearly inform the readers on how different treatments reduced regeneration, or what the ‘with the cryovial’ treatment was
Line 40 – keenly short-lived seeds? Definition of keenly is “in an eager or enthusiastic manner”. I do not think the seeds are eager to be short-lived. Line 93 – “glowing velocity”? There are a few of these kinds of typographical errors throughout the manuscript that the authors need to fix.
Line 73 – reference 17 listed but should be 18?
Line 83 – “This is the first investigation into the relevance of OAs, as a signature or oxidative stress …” I do not think all OAs are necessarily a signature of oxidative stress, some certainly but not all the OAs assessed in the manuscript.
Line 108 – The authors use the term mono-basic, di-basic etc here, but later in the manuscript use one-component, two-component etc. Need to consistently stick to one term, preference for one-component style as monobasic and dibasic are commonly associated with acids.
Line 122 – The OA content was correlated with CPA concentration; it was not listed if the reported OA content was in ng/2mg fresh weight or ng/2mg dry weight. Would be interesting to see if this correlation holds to the water content in the shoot tips too. It is also strange to normalise the OA to ng/2mg instead of just ng/mg
Line 214 – why wasn’t this data presented? This has resulted in a strange empty spot in figure 5.
Line 361 – avoid using the term ‘etc.’ It is ambiguous what other amino acids decreased after cryopreservation in that study when this is done.
Line 422 to 435 – the authors refer to a lot of medical literature here, would be preferable to discuss these OAs in the context of plants’ stress tolerance before referring to the medical literature.
Line 519 – software used to analyse the GC-MS data.
Star graphs – due to the huge fold changes in 2-hydroxybutyric acid a lot of the comparison in these graphs are very difficult to see. Additionally, these graphs would look very different if graphed of the amount instead of the normalised values, an interesting comparison to make. Alternatively, the graphs could be split into their groups as outline at line 394 so the more subtle differences would show up for groups 2 and 3.
Figure 7 is listed twice.
Figure 3, 5 and 7. Legends are incomplete – list surv and rege instead of survival and regeneration. LN regeneration missing completely from fig 5. The figure captions do not state what the error bars are (or n)
Figure 9 (listed figure 7). The y axis labels are unclear what they are referring to.
See above
Reviewer 2 Report
Reviewer comments on the manuscript
Manuscript Number: plants-2508554
Title: Metabolomic Dynamics of Organic Acids During the Droplet-Vitrification Procedure Can Be a Signature of Oxidative Stress in Endangered Species Pogostemon yatabeanus
Article Type: Article
Title
The title is quite clear and well defined, but I don't think the term "endangered species" is important enough to be in the title. I would move it to the abstract section.
Abstract
The abstract is informative and well-written. I suggest moving the abbreviation "OAs" from line 24 to line 23 when the term first appeared. I suggest replacing the abbreviation "CORREL" with "r" on line 26 and also in the rest of the article.
Keywords
Keywords are appropriate.
Introduction
Introduction part is clear, the objectives are well defined.
Results
The results are well structured and quite clearly presented. Minor edits please. I assume that the information about "unloader (UL)" is missing in the second sentence of the first paragraph (line 92).
Discussion
This section is clear and well prepared. Please correct the references to Figure 9 in the text.
Material and Methods
This section is generally clear and well structured, but some corrections are needed. Use term “stress condition” instead of “stresse conditions” at the line 473. The last part of this section (4.4. Recovery Assessment and Statistical Analysis”) should be improved. I could not find information on the calculation of the correlation coefficient. I understand that the only data on explant recovery were statistically analyzed. So, statistically significant differences were not evaluated for OAs content?
Tables and Figures
I have a few suggestions to improve this section. The tables are quite clear and well designed, but sometimes it is difficult to recognize individual numbers due to the large number of decimal places. Therefore, I recommend rounding the numbers. Two decimal places are redundant for hundreds, any decimal places are unnecessary for thousands or higher. Rounding numbers does not have the same principle for a similar number in different rows. The character legends are rather brief. I would have expected more information about the data explaining the numbers: the number of replicates, what the vertical bars represent (standard errors of the mean?). I would properly describe the variants. I believe there was a typo in the legend for Figure 1. I would expect 10% MSF instead of 3% MSF in line 102. Or am I wrong? One dot at the end of the legend to Figure 1 is redundant. Replace "Figure 9." instead of "Figure 7." on line 404 and check the links in the text.
References
References cited are mostly recent and relevant publications. The list does not contain an excessive number of self-citations.
Overall evaluation
The topic of the manuscript falls within the general scope of the journal. The manuscript brings novel results. Correct interpretations and conclusions are justified by the data and are consistent with the objectives. The manuscript is scientifically sound, and the experimental design is appropriate to test the hypothesis. The results of the manuscript are reproducible based on the details given in the method section. Figures/tables/ need minor corrections. The English is correct and understandable for a multidisciplinary and multinational readership. The manuscript is acceptable for publication after minor revision.
Author Response
I apologize for the miscommunication. The revised manuscript did not reach reviewers.
Hopefully, you can see an entirely revised manuscript.

Reviewer 3 Report
The manuscript entitled "Metabolomic Dynamics of Organic Acids During the Droplet Vitrification Procedure Can Be a Signature of Oxidative Stress in Endangered Species Pogostemon yatabeanus", from the authors Hyoeun Lee, Byeongchan Choi, Songjin Oh, Hana Park, Elena Popova, Man-Jeong Paik and Haeng-Hoon Kim.
The manuscript is interesting, well conceived, the experimental results are clearly presented and analyzed. This manuscript is part of a wider research and will contribute to the study in this field.
A table is shown in the text (lines 477 to 485) and mentioned in the text of the manuscript (line 474). The table is labeled "Table 3" and should be "Table 5". I ask the authors to correct the error in the manuscript.
Figure 5 (line 217) shows 4 different bars on the graph and only three in the legend. I ask the authors to clarify what the fourth blue bar on the graphic means?
The manuscript mentions the abbreviation GA3 for gibberellic acid (lines 148, 200, 242, 268, 304, 467, 482 and 483). However, in the text of the manuscript there is also the abbreviation GA for which there is no full name (lines 254, 255 and 265). It is unclear whether it is one or two different substances. It might be a mistake. I ask the authors to make a correction in the manuscript.
In "template.dot" for the journal "Plants" there is a section "5. Conclusions". There is no part "5. Conclusions" in this manuscript. I ask the authors to add "5. Conclusions" to their manuscript.
I consider that manuscript should be published in journal „Plants“ after minor corrections.
Author Response
I apologize for the miscommunication. The revised manuscript did not reach the first round.

Reviewer 4 Report
The study explores cryopreservation's unique potential at -196°C for conserving threatened plant species with limited seed availability. A droplet-vitrification (DV) technique was meticulously developed, involving sequential stages: preculture, osmoprotection, and cryoprotection with alternative PVS A3-80%. The procedure achieved 90% regeneration using the standard condition, contrasting with 28% using a cryovial. Gas chromatography-tandem mass spectrometry revealed dynamic changes in organic acids (OAs) throughout DV, linked to cryoprotectant concentration. The substantial increase of select OAs during cooling and rewarming, particularly lactic and glycolic acids, highlights stress-induced metabolic responses aiding in countering cryoprotectant-associated challenges. The potential role of oxidative stress-related biomarkers in plant cryopreservation warrants further investigation. The article is interesting and valuable. Nonetheless, there are some drawbacks that need to be corrected before publication.
Major issues:
- Tables and graphs lack statistical symbols. Therefore, it is impossible to verify the description of the results in the main body of the text.
- Standard deviations in Tables are missing.
- Conclusions are missing.
Minor issues:
- All abbreviations must be explained when mentioned for the first time, e.g. line 74 – what is Al?
- Line 80: do not repeat the explanation of abbreviations?
- Figure 1: scale bars are missing.
- Unit style is incorrect. It should be ng·mg-1 (not ng/mg)
- What was the light source?
- What time after thawing was the biochemical analysis performed?
- What was the design of the experiment? CRD?
- In the Reference list, follow the MDPI formatting style. DOI numbers are missing.
- I suggest updating some older references at the beginning of the Introduction.
Minor grammatical and punctuation errors require correction.
Round 2
Reviewer 1 Report
The revised manuscript (v2) does not show the changes listed by the authors in the response to reviewer comments. E.g. ">> Star graphs are removed. Alternatively, we added heat maps of the normalized OA values in the Results for a better visual representation of the OA content data." the manuscript still shows the star graphs and no heat maps in the pdf supplied to the reviewers.
However, based on the authors comments regarding the experimental design of this study: the lack of replication for the OA analysis and choosing to only show results from one of the replicates is insufficient for publication. Based on the comment to reviewer 4 "However, the absolute numbers in ng were sometimes quite different between the two biological replications, probably because of the physiological difference in the starting in vitro plant material" indicates that OA content is quite variable. Increased replicates for the OA content (n = 5 or more) is probably require for proper statistical analysis to identify relevant significant changes in OA due to the cryopreservation process.
"The primary reason for missing additional replication may be burdensome for material preparation. The shoot tips for LN regeneration and OA analysis were no less than 2,650 (regeneration 950, OA 1,700) from the subculture, nodal section culture, and excision of shoot tips to the cryopreservation and OA analysis. It’s a huge work!" - I agree that this is a very large study, but ensuring sufficient replicates in the experimental design stage is vital to ensure valid results are obtained for publication.
Authors have addressed the suggestions from the first round regarding quality of english language
Author Response
I apologize for the miscommunication. The revised manuscript did not reach the reviewers in the first round.
Now, you can see the entirely revised manuscript.

Reviewer 4 Report
The authors did not include my suggestions/corrections in the revised manuscript.
English is fine
Author Response
I apologize for the miscommunication. The revised manuscript did not reach reviewers in the first round.
Hopefully, you can see the entirely revised manuscript.

Round 3
Reviewer 1 Report
Based on the authors comments regarding the experimental design of this study: the lack of replication for the OA analysis and choosing to only show results from one of the replicates is insufficient for publication. Based on the comment to reviewer 4 "However, the absolute numbers in ng were sometimes quite different between the two biological replications, probably because of the physiological difference in the starting in vitro plant material" indicates that OA content is quite variable. Increased replicates for the OA content (n = 5 or more) is probably require for proper statistical analysis to identify relevant significant changes in OA due to the cryopreservation process.
"The primary reason for missing additional replication may be burdensome for material preparation. The shoot tips for LN regeneration and OA analysis were no less than 2,650 (regeneration 950, OA 1,700) from the subculture, nodal section culture, and excision of shoot tips to the cryopreservation and OA analysis. It’s a huge work!" - I agree that this is a very large study, but ensuring sufficient replicates in the experimental design stage is vital to ensure valid results are obtained for publication.
Reviewer 4 Report
The caption to Figure 2 is impossible to read. The statistical symbols are misplaced.